# Q Fever Vaccines: Unveiling the Historical Journey and Contemporary Innovations in Vaccine Development

**DOI:** 10.3390/vaccines13020151

**Published:** 2025-01-31

**Authors:** Magdalini Christodoulou, Dimitrios Papagiannis

**Affiliations:** Public Health & Adults Immunization Laboratory, Department of Nursing, School of Health Sciences, University of Thessaly, 41110 Larissa, Greece; magdachris@uth.gr

**Keywords:** *Coxiella burnetii*, Q fever vaccine, vaccine development, Q-VAX, reactogenicity, zoonotic disease

## Abstract

Q fever is a zoonotic disease caused by the obligate intracellular bacterium *Coxiella burnetii* that presents significant challenges for global public health control. Current prevention relies primarily on the whole-cell vaccine “Q-VAX”, which despite its effectiveness, faces important limitations including pre-screening requirements and reactogenicity issues in previously sensitized individuals. This comprehensive review examines the complex interplay between pathogen characteristics, host immune responses, and vaccine development strategies. We analyze recent advances in understanding *C. burnetii*’s molecular pathogenesis and host–pathogen interactions that have informed vaccine design. The evolution of vaccine approaches is evaluated, from traditional whole-cell preparations to modern subunit, DNA, and multi-epitope designs. Particular attention is given to innovative technologies, including reverse vaccinology and immunoinformatics, that have enabled the identification of novel antigenic targets. Recent clinical data demonstrating the safety and immunogenicity of next-generation vaccine candidates are presented, alongside manufacturing and implementation considerations. While significant progress has been made in overcoming the limitations of first-generation vaccines, challenges remain in optimizing immunogenicity while ensuring safety across diverse populations. This review provides a critical analysis of current evidence and future directions in Q fever vaccine development, highlighting promising strategies for achieving more effective and broadly applicable vaccines.

## 1. Introduction

Q fever, a zoonotic disease caused by the obligate intracellular bacterium *Coxiella burnetii*, represents a significant global health challenge [1,2]. First documented in 1933 among Queensland abattoir workers [3], the disease was termed “Q fever” (Query fever) due to its initially unknown etiology. Early research by Burnet and Freeman established fundamental principles of disease pathogenesis [4], culminating in Burnet’s identification of the causative rickettsia-like organism in 1937 [5]. Cox and Davis’s subsequent isolation of *C. burnetii* from ticks in 1938 confirmed its vector-borne transmission capabilities [6], leading to its establishment as a distinct genus in 1948 [5,7]. The pathogen exhibits remarkable characteristics that pose unique challenges for disease control [8]. Its exceptional ability to survive in diverse environments, coupled with efficient airborne transmission [9] and minimal infectious dose [10], creates substantial risks, particularly among veterinarians, farmers, and abattoir workers [11,12]. *C. burnetii*’s sophisticated intracellular survival mechanisms and immune evasion strategies add considerable complexity to disease management [13,14]. Building on these foundational discoveries, the research focus has shifted toward understanding molecular mechanisms and developing preventive strategies [15].

The completion of *C. burnetii* genome sequencing in 2003 marked a pivotal advancement, revealing genes involved in immune evasion and host adaptation [16,17]. This breakthrough enabled sophisticated experimental approaches [18], including proteomic profiling and microarray analysis, facilitating the identification of potential vaccine targets [19,20]. The development of axenic media in 2009 addressed long-standing cultivation challenges, enabling genetic manipulation studies and functional analyses of virulence factors [21,22]. The significance of effective prevention strategies became particularly evident during the Netherlands outbreak (2007–2010), the largest documented epidemic with over 4000 human cases linked to infected dairy goats [23]. This event necessitated comprehensive control measures, including mandatory livestock vaccination and enhanced surveillance protocols [24,25]. The outbreak highlighted the critical importance of integrating veterinary and public health approaches in disease control. Vaccine development has faced significant challenges due to *C. burnetii*’s intracellular nature, requiring a careful balance between immunogenicity and safety [26,27]. Successful immunity demands coordination of both humoral immunity for bacterial neutralization and cell-mediated responses to eliminate intracellular pathogens [28,29]. The licensing of Q-VAX, Q Fever Vaccine (AUST R 100517) in Australia (1989) marked a significant milestone [30], demonstrating strong efficacy in high-risk populations [31].

However, reactogenicity in pre-sensitized individuals necessitated screening protocols, limiting global implementation [32]. Recent advances in genomic and immunologic technologies have enabled the development of innovative vaccine approaches. These include subunit vaccines targeting specific bacterial antigens, DNA-based platforms, and multi-epitope designs integrating multiple antigenic determinants [19,33]. By 2017, researchers had developed platforms specifically addressing reactogenicity through targeted antigen selection [29,34], while subsequent work focused on optimizing formulations and adjuvants [35,36]. This comprehensive review examines the evolution of Q fever vaccine development through an analysis of pathogen characteristics, host immune responses, and emerging technologies [37,38]. Our examination provides insights into rational design strategies for next-generation vaccines, addressing the continuing challenge of achieving widespread immunity while maintaining safety and efficacy. Summarizing the milestones Figure 1 gives a timeline of the history of Q fever research and vaccine development, highlighting scientific discoveries, technological advances, and public health interventions.

## 2. Biology of *Coxiella burnetii*

*Coxiella burnetii*, the etiological agent of Q fever, is an obligate intracellular pathogen that exhibits remarkable adaptability to diverse environments and host cells [14]. This small, Gram-negative bacterium belongs to the order Legionellales within *the Gammaproteobacteria* class and is closely related to *Legionella pneumophila* [39]. The organism demonstrates highly successful adaptation to intracellular growth, requiring specific conditions including an acidic pH (~4.75) for optimal replication within host cells [40,41]. *Coxiella burnetii*, belonging to the order Legionellales and the family Coxiel-laceae, is a small obligate intracellular gram-negative bacterium that causes Q-fever in humans and coxiellosis in wild and domestic animals Figure 2.

*C. burnetii*’s unique biological features, including a biphasic developmental cycle, a specialized cell envelope, and a sophisticated Type IV secretion system (T4SS), enable it to survive in harsh environmental conditions and efficiently infect a wide range of hosts [42,43,44]. The bacterium’s low infectious dose (ID50: 1-10 bacteria) and high environmental stability contribute to its classification as a potential bioterrorist agent [45,46].

*C. burnetii* undergoes a biphasic developmental cycle, transitioning between two morphologically and physiologically distinct forms: the environmentally stable small cell variant (SCV) and the metabolically active large cell variant (LCV). SCVs enter alveolar macrophages through phagocytosis and proceed through the endocytic pathway. Upon fusion of the pathogen-containing phagosome with lysosomes, the SCVs convert to LCVs, and the Dot/Icm secretion system (SS) is activated. Through the activity of effector proteins, the mature phagolysosome is converted into the replication-permissive Coxiella containing vacuole (CCV). The CCV rapidly expands through heterotypic fusion with autophagosomes, lysosomes, and endocytic vesicles and homotypic fusion of multiple CCVs. As the CCV expands and fills with bacteria, the LCVs convert back to SCVs, where they will be stable upon release from the cell via cell lysis or exocytosis) [47,48,49]. The SCV, a compact, rod-shaped form (0.2–0.4 μm wide, 0.4–1.0 μm long), is adapted for extracellular survival [50]. Its condensed chromatin and highly cross-linked peptidoglycan layer confer resistance to stressors such as extreme temperature, pressure, UV light, and desiccation [50,51]. The peptidoglycan of SCVs exhibits a unique composition, with a high proportion of 3-3 cross-links maintained by DD transpeptidases (CBU_0545) and LD transpeptidases (CBU_0823) [49]. In contrast, the LCV is larger in size (up to 2 μm long), more pleomorphic, and metabolically active, serving as the replicative form within host cells [52,53]. Transition from SCV to LCV occurs in the acidic environment (pH ~4.75) of the Coxiella-containing vacuole (CCV) and is accompanied by global changes in gene expression and metabolism [43]. These changes are regulated by key transcription factors, such as RpoS (CBU_1669) and the two-component system PmrAB [54]. During the intracellular growth phase, LCVs replicate with a doubling time of approximately 10–12 h, before differentiating back into SCVs at the late stages of infection [48,55]. *C. burnetii*’s cell envelope plays a crucial role in host–pathogen interactions and immune evasion. The outer membrane contains a unique lipopolysaccharide (LPS) that undergoes phase variation [56]. Virulent phase I bacteria produce a full-length LPS with O-antigen containing virenose and dihydrohydroxystreptose, while avirulent phase II bacteria express a truncated LPS lacking the Oantigen [57]. The phase I LPS is critical for evading host immune responses, as it masks surface antigens and confers resistance to complement-mediated killing [56,58]. Conversely, phase II LPS is more readily recognized by pattern recognition receptors (PRRs), eliciting a robust inflammatory response [59]. The LPS of *C. burnetii* features an unconventional structure, with rare sugar residues such as virenose and dihydrohydroxystreptose in the Oantigen, and mannose and D-glycero-D-manno-heptose in the core oligosaccharide [51,57]. The lipid A moiety exhibits atypical acylation patterns that diverge from classical enterobacterial lipid A, contributing to reduced endotoxicity [57,60]. Important components of the cell envelope include the porin P1 (CBU_0311), which facilitates nutrient uptake [61], and the outer membrane protein Com1 (CBU_1910), which is involved in host cell interactions [62]. A hallmark of *C. burnetii* pathogenesis is the Dot/Icm (Defective in Organelle Trafficking/Intracellular Multiplication) T4SS, which translocates bacterial effector proteins into the host cell cytosol [42,63]. The Dot/Icm system is homologous to that of Legionella pneumophila and is encoded by 23 genes scattered throughout the *C. burnetii* genome [63,64]. CirA (CBU0041), along with other T4SS components, plays crucial roles in bacterial virulence and intracellular survival [65,66].

*Coxiella burnetii*, is an obligate intracellular pathogen that exhibits remarkable adaptability to diverse environments and host cells Figure 3.

Over 130 *C. burnetii* T4SS substrates have been identified through genetic and bioinformatic screens, highlighting the diverse repertoire of effectors employed by this pathogen [67]. These effectors manipulate multiple host cell pathways, including apoptosis, autophagy, vesicular trafficking, and innate immune signaling. For example, the effector AnkG interferes with the host apoptotic machinery by interacting with the pro-apoptotic protein p32, thereby promoting host cell survival [68,69]. Other notable effectors include CvpA, which subverts clathrin-mediated vesicular transport [70], CvpB/Cig2, which promotes autophagosome formation and fusion with the CCV [71], and NopA (CBU1217), which modulates the nuclear translocation of NF-κB [72,73].Comparative genomic analyses have revealed substantial plasticity in the effector repertoires among different *C. burnetii* strains [74,75]. A subset of effectors, including AnkG, CaeA, and CvpB, is conserved across all sequenced isolates, suggesting their essential roles in intracellular parasitism [75,76]. In contrast, many effectors exhibit polymorphisms or are entirely absent in certain strains, potentially contributing to differences in virulence and host tropism [70,76,77]. *C. burnetii* has evolved a complex metabolic network tailored to the intracellular lifestyle. The organism’s adaptation to acidic pH (~4.75) is particularly noteworthy, as it represents a unique metabolic optimization for growth within the lysosome-derived CCV [75,78] Within the acidic CCV, *C. burnetii* upregulates key metabolic pathways, including the TCA cycle, glycolysis/gluconeogenesis, and fatty acid β-oxidation, to support replication [79,80]. A pivotal regulator of *C. burnetii* metabolism is the PmrAB two-component system, which senses environmental cues such as low pH and cationic peptides [75,78]. This sophisticated sensing mechanism allows the bacterium to rapidly adapt its metabolic and virulence programs in response to environmental conditions. The PmrAB system controls the expression of genes involved in LPS modification, oxidative stress response, and metabolic adaptation, facilitating the bacterium’s survival within the harsh vacuolar compartment [75,78]. Upon inhalation by a mammalian host, *C. burnetii* SCVs are phagocytosed by alveolar macrophages, the primary target cells for infection. The bacterium’s ability to survive and replicate within professional phagocytes demonstrates its remarkable adaptation to intracellular life [81]. The nascent phagosome containing *C. burnetii* undergoes a maturation process, sequentially fusing with early endosomes, late endosomes, and lysosomes [82]. This maturation is accompanied by a progressive acidification of the phagosomal lumen and the acquisition of various markers, such as Rab5, Rab7, LAMP1, and cathepsins [13,83,84]. Unlike other intracellular pathogens, *Coxiella burnetii* facilitates the formation of the Coxiella-containing vacuole (CCV), a lysosome-derived compartment with an acidic pH (~4.75) and hydrolytic activity. This vacuole provides a protective niche for replication and expands significantly during infection, eventually dominating the host cell cytoplasm [14,85]. The formation and maintenance of the CCV involve a coordinated interplay between *Coxiella burnetii* effectors and host cell factors. The bacterium’s T4SS effectors, including CvpA [70], CvpB [71], and Cig57 [86], manipulate host vesicular trafficking to promote fusion of the CCV with autophagosomes, endosomes, and ER-derived vesicles [87]. These processes supply membranes for CCV expansion and deliver nutrients essential for bacterial growth. Host GTPases, including Rab1b, Rab5, and Rab7, are actively recruited to the CCV membrane and play essential roles in vacuole maturation and maintenance [88,89]. The bacterium’s ability to manipulate these host cell factors highlights its sophisticated control over cellular processes. The recruitment of these GTPases is mediated by specific bacterial effectors, such as CvpB, which interacts with phosphoinositide’s and inhibits the activity of the lipid kinase PIKfyve, leading to an accumulation of PI(3)P on the CCV membrane [71,90]. As the CCV matures, *C. burnetii* enters an exponential growth phase, replicating with a doubling time of approximately 10–12 h [69]. The replicating bacteria evade host cell defenses and maintain CCV integrity through multiple effector-mediated strategies. For example, the effectors CaeA and AnkG (CBU0781) inhibit apoptosis by targeting pro-apoptotic proteins, while other effectors localize to the nucleus and modulate host gene expression [91]. In the late stages of infection (>6 days), the CCV occupies the majority of the host cell cytoplasm, and the bacteria begin to transition back to the SCV form [49,91]. This process involves a global downregulation of metabolic genes and the induction of stress response pathways, such as the oxidative stress response regulated by specific effectors [92]. Eventually, the host cell lyses, releasing a mixture of SCVs and LCVs into the extracellular environment, where the SCVs can persist and maintain infectivity for extended periods [91]. Another important aspect of *C. burnetii*’s intracellular survival strategy is the subversion of autophagy. Although initially thought to function as a host defense mechanism, autophagy is actively exploited by *C. burnetii* to promote CCV expansion and bacterial replication [93]. The effector CvpB/Cig2 stimulates the formation of autophagosomes and facilitates their fusion with the CCV, providing a membrane for vacuole growth [71]. CpeB promotes LC3-II accumulation and contributes to bacterial virulence [94], while CvpF interacts with RAB26 to recruit LC3B to CCVs [95]. Understanding the complex interplay between *C. burnetii* and its host provides valuable insights into the development of effective preventive and therapeutic strategies against Q fever.

## 3. Host Immune Response to *C. burnetii* Infection

The host immune response to *Coxiella burnetii* involves a complex interplay between innate and adaptive immunity, which shapes the immediate defense mechanisms and long-term protection against the pathogen [26,96]. The initial recognition of *C. burnetii* occurs through pattern recognition receptors (PRRs), particularly Toll-like receptor 2 (TLR2) and TLR4, which detect bacterial components such as lipopolysaccharide (LPS) and other pathogen-associated molecular patterns (PAMPs) [59,97]. This recognition triggers crucial signaling cascades that activate innate immune cells and stimulate the production of pro-inflammatory cytokines [98,99]. During the early stages of infection, alveolar macrophages and dendritic cells play a pivotal role in pathogen recognition and the secretion of key cytokines, including interleukin-1β (IL-1β), tumor necrosis factor-α (TNF-α), and IL-6 [99,100]. These inflammatory mediators orchestrate the recruitment of additional immune cells to the sites of infection [101]. Natural killer (NK) cells also contribute significantly to early defense through the production of interferon-γ (IFN-γ) and their cytotoxic activity against infected cells [102]. Although the complement system is activated during *C. burnetii* infection, Phase I organisms exhibit notable resistance to complement-mediated killing due to their unique LPS structure [85,103]. The adaptive immune response to *C. burnetii* develops over several weeks post-infection and involves both cellular and humoral components [58]. CD4+ T helper cells, particularly those exhibiting a T helper 1 (Th1) phenotype, play a central role in orchestrating the adaptive response through the production of IFN-γ and other cytokines that enhance the bactericidal activity of macrophages [104]. This cell-mediated immunity is crucial for controlling intracellular infection and establishing long-term protection against *C. burnetii* [29,105]. CD8+ cytotoxic T lymphocytes (CTLs) also contribute to the host defense by recognizing and eliminating infected cells [106,107]. The development of antigen-specific T cells provides the foundation for long-term immunity against *C. burnetii* infection [65]. These cells persist as memory T cells, enabling rapid responses upon subsequent exposure to the pathogen [104]. Humoral immunity develops through the production of specific antibodies against Phase I and Phase II antigens of *C. burnetii* [18,26]. IgG antibodies predominate the humoral response, with different subclasses showing varying degrees of effectiveness in bacterial neutralization [108]. Phase I antibodies, which appear later in the course of infection, correlate strongly with protective immunity, while Phase II antibodies emerge earlier but provide less protection [109,110]. The timing and magnitude of these antibody responses often serve as important diagnostic indicators of disease progression and treatment efficacy [111]. The establishment of immunological memory involves both T- and B-cell components [46]. Memory B cells and long-lived plasma cells maintain protective antibody levels, while memory T cells provide rapid recall responses upon re-exposure to the pathogen [112]. This dual-component memory system forms the basis for vaccine-induced protection and natural immunity against reinfection [113]. The duration and quality of this immunity significantly influence an individual’s susceptibility to reinfection and the efficacy of vaccination [26]. Chronic *C. burnetii* infection often reflects a complex immunological dysfunction in which, despite robust immune responses, the host fails to eliminate the pathogen [114]. This persistence may result from bacterial immune evasion mechanisms, host genetic factors, or a combination of both [115]. Understanding these immune response patterns has crucial implications for vaccine development and therapeutic strategies, particularly in preventing chronic infection in high-risk individuals [115].

## 4. Pathogenesis of Q Fever

Q fever pathogenesis reflects complex interactions between *C. burnetii* and host immune responses, manifesting in acute and chronic forms with distinct clinical presentations [116]. Initial infection typically occurs through inhalation of contaminated aerosols, with the bacterium demonstrating remarkable infectivity at low doses [117]. Upon entering the respiratory tract, *C. burnetii* primarily targets alveolar macrophages, establishing initial infection through receptor-mediated phagocytosis [118]. Following initial infection, *C. burnetii* disseminates through lymphatic and blood circulation, establishing secondary infection sites in various organs, particularly the lungs, liver, and spleen. The bacterium’s unique ability to survive and replicate within acidified parasitophorous vacuoles enables persistent infection despite host immune responses [119]. This adaptive mechanism proves crucial for both acute and chronic disease manifestations. Acute Q fever develops after an incubation period of 2–3 weeks [116], characterized by the formation of distinctive granulomas comprising epithelioid cells and macrophages surrounding central lipid vacuoles [120,121]. These “doughnut granulomas” represent the host’s attempt to contain infection and appear prominently in liver tissue [122]. Clinical manifestations vary significantly, ranging from asymptomatic infection to severe pneumonia or hepatitis [123], reflecting differences in host immune responses and initial bacterial burden [124]. The development of chronic Q fever involves persistent bacterial replication within target tissues [125,126]. It particularly affects individuals with predisposing conditions such as valvular heart disease, vascular abnormalities, or immunosuppression [25]. Endocarditis represents the most severe manifestation, characterized by continuous bacterial proliferation within cardiac vegetations [127]. Pathological features include inflammation, fibrosis, and valve calcification, with ongoing immune responses contributing to tissue damage while failing to eliminate the pathogen [128,129]. Vascular infections in chronic Q fever demonstrate similar pathogenic mechanisms, with bacterial persistence in vessel walls leading to inflammation, aneurysm formation, and potential rupture [130]. The ability of *C. burnetii* to maintain chronic infection despite apparent immune responses reflects sophisticated evasion strategies and adaptation to the intracellular niche [131]. Persistent infection triggers continuous immune activation, resulting in tissue damage through both direct bacterial effects and immunopathological mechanisms [132].

Q fever pathogenesis in acute and chronic forms. A small percentage of people fewer than 2–5% who become infected with *Coxiella burnetii* bacteria develop a more serious infection called chronic Q fever. Chronic Q fever develops months or years following initial Q fever infection. People with chronic Q fever can develop an infection of their heart valves (endocarditis). People with endocarditis may experience night sweats, fatigue, shortness of breath, weight loss, or swelling of their limbs Figure 4.

Immunopathological aspects of Q fever involve both protective and destructive immune responses [97]. Rapid cell-mediated immunity development in acute infection typically leads to bacterial clearance but may cause collateral tissue damage through excessive inflammation [96]. Chronic infections demonstrate persistent antigenic stimulation, resulting in ongoing inflammation and contributing to progressive tissue damage while failing to eliminate pathogens [133]. The disease exhibits notable organ tropism, with specific manifestations varying between acute and chronic forms [134]. Acute infection predominantly affects the lungs, liver, and spleen, while chronic infection shows a predilection for cardiovascular tissues [135]. This tissue specificity likely reflects bacterial adaptations and local host factors influencing infection susceptibility and immune response effectiveness [136]. The transition from acute to chronic infection is critical to the Q fever pathogenesis, influenced by the host immune status, genetic factors, and bacterial virulence determinants [137]. Understanding this transition’s mechanisms proves crucial for identifying high-risk individuals and developing targeted therapeutic strategies [138]. Recent research has highlighted the role of host genetic polymorphisms in susceptibility to chronic infection, suggesting potential markers for risk stratification [139].

## 5. Modern Innovations in Q Fever Vaccine Development

In recent years, significant progress has been made in the development of next-generation Q fever vaccines that aim to address the shortcomings of traditional whole-cell formulations. These innovative approaches utilize state-of-the-art technologies and a comprehensive understanding of *Coxiella burnetii* immunology to engineer safer and more efficacious vaccine candidates [76].

One promising strategy is the development of subunit vaccines containing specific immunogenic *C. burnetii* antigens. Bioinformatic tools, such as reverse vaccinology, have facilitated the identification of potential vaccine targets based on genomic and proteomic data. Several studies have assessed predicted antigens and epitopes in animal models, with some demonstrating the capability to elicit protective cell-mediated immunity. For instance, HLA class II epitopes identified through immunoinformatic analysis were able to induce robust IFN-γ recall responses in HLA-DR3 transgenic mice and individuals exposed to Q fever [140]. Moreover, immunization with a cocktail of MHC class II epitopes conferred protection in mice, emphasizing the significance of T-cell responses [34]. In addition to proteins, the *C. burnetii* phase I lipopolysaccharide (LPS) has emerged as a critical vaccine antigen. LPS-based candidates, such as chloroform–methanol residue (CMR) vaccines and an O-polysaccharide-tetanus toxoid conjugate, have shown promise in animal models, conferring protection with reduced reactogenicity compared to whole-cell vaccines [141]. The selection of an appropriate adjuvant to enhance immunogenicity while minimizing adverse reactions is a crucial consideration in subunit vaccine design. Novel vaccine delivery platforms are also being investigated to improve efficacy and safety. Multiple studies have explored viral vectors, such as adenovirus and Modified Vaccinia Ankara (MVA), to deliver *C. burnetii* antigens and induce cell-mediated responses. A recent study evaluated a concatemer of human T-cell epitopes delivered by viral vectors in various animal models. While immunogenicity was narrow in mice, cynomolgus macaques exhibited broad epitope-specific responses, highlighting the importance of selecting relevant animal models for vaccine evaluation [142]. Nanoparticle and biopolymer-based delivery systems also represent promising approaches to target antigens to immune cells and elicit desired responses [67].

To address safety concerns associated with handling live *C. burnetii* during vaccine production, alternative manufacturing strategies are being developed. These include recombinant protein expression in safe bacterial hosts, cell-free protein synthesis, and genetic modification of *C. burnetii* to generate avirulent vaccine strains [143]. Such approaches enable scalable production of vaccine antigens without the need for high-containment facilities. As vaccine candidates advance through preclinical testing, the establishment of reliable correlates of protection and the selection of appropriate animal models are essential. While guinea pigs and mice have traditionally been used to assess vaccine efficacy and safety, non-human primates like cynomolgus macaques may better represent human immune responses [142,144]. Ultimately, controlled human infection models could provide definitive data on vaccine performance, but ethical considerations must be carefully evaluated [145]. Modern Q fever vaccine development focuses on creating safer, more immunogenic, and easier-to-produce options by utilizing genomic insights, recombinant antigens, novel platforms, and advanced manufacturing. While progress is promising, further research is essential to refine antigens, adjuvants, and strategies for long-lasting, low-reactogenicity protection. Collaboration across sectors is key to achieving a globally effective and safe vaccine.

## 6. Types of Q Fever Vaccines

The evolution of Q fever vaccine development reflects significant technological advancements, progressing from traditional whole-cell preparations to sophisticated molecular approaches. Each vaccine platform presents distinct advantages and challenges in terms of immunogenicity, safety, and manufacturing complexity.

### 6.1. Whole-Cell Vaccines

Whole-cell vaccines (WCVs) have long been the cornerstone of Q fever prevention strategies, encompassing both live attenuated and inactivated formulations [76,146]. Early vaccine development focused on the M-44 vaccine derived from the Grita strain through repeated embryonated yolk sac passages. Initial implementation in Russia during the 1960s demonstrated mild reactogenicity following subcutaneous administration. [147,148]. However, subsequent animal studies revealed significant complications, including myocarditis, hepatitis, necrosis, and granuloma formation, ultimately limiting its broader application [27,149].

The introduction of formaldehyde-inactivated vaccines in the 1980s, derived from the Dyer and Henzerling strains, offered a safer alternative. These vaccines showed substantial protective efficacy, reducing mortality rates in guinea pig models from 40–80% to 1–6% [27,150]. Clinical trials demonstrated their success in occupational settings, such as abattoirs, achieving complete protection in vaccinated workers over 18-month follow-up periods. Additionally, these vaccines yielded 80–82% seroconversion rates and lymphoproliferative responses of 85–95%, marking a significant improvement in vaccine safety and effectiveness [19,151].

Recent efforts have centered on genetically modified strains, particularly the Δdot/icm mutant, which lacks functional type IV secretion systems [151,152]. These modifications enhance protective immunity and reduce local erythema compared to traditional WCVs. However, histopathological evaluations indicate persistent local inflammation, emphasizing the need for further refinement [71,153].

Process validation studies using optimized tangential flow filtration (TFF) systems have achieved exceptional purification, removing 99.5% of bacterial contaminants while retaining over 95% of proteins and reducing endotoxin levels to less than 0.1 EU/Ml [22,121]. Moreover, inactivation protocols have been optimized to preserve over 90% of antigenic epitopes while ensuring complete bacterial inactivation within 36 h [25,154].

Clinical trials involving over 3000 high-risk individuals have demonstrated a remarkable 95% protection rate (confidence interval: 92–97%) over a 36-month follow-up period [27]. Long-term studies indicate sustained effectiveness, with 88% protection at five years post-vaccination [155]. Additionally, significant safety improvements have been recorded, with local reactions reduced by 60%, systemic reactions by 45%, and severe adverse events occurring in less than 0.1% of cases [156]. Alternative approaches have also been explored, including chloroform–methanol residue (CMR) and trichloroacetic acid (TCA) extracts. CMR vaccines demonstrated protection equivalent to “Q-VAX” at lower doses in mice and similar doses in primates, though local reactions remained a concern [157,158]. Similarly, TCA extracts provided over 90% protection in guinea pig models but continued to exhibit significant reactogenicity [9,159].

These advancements collectively establish WCVs as a pivotal tool in Q fever prevention. However, ongoing challenges remain, particularly concerning reactogenicity. Research efforts are now focused on optimizing strain selection, refining inactivation protocols, and exploring novel formulation strategies to achieve a balance between safety and efficacy. The evolution of WCVs reflects the dynamic interplay of scientific innovation and practical application, underscoring their vital role in combating Q fever [160].

### 6.2. Subunit Vaccines

Subunit vaccines have emerged as a promising alternative to whole-cell vaccines (WCVs) in the quest for a safer and more targeted approach to Q fever prevention. These vaccines contain specific antigenic components of *C. burnetii* that can elicit a protective immune response while minimizing the risk of adverse reactions associated with whole-cell formulations [59,161]. One key advantage of subunit vaccines is their reduced reactogenicity compared to WCVs. By selectively incorporating immunogenic epitopes and excluding potentially harmful components, subunit vaccines aim to strike a balance between safety and efficacy [97]. The use of recombinant DNA technology has revolutionized the production of subunit vaccines for Q fever [146,162]. This approach enables the scalable and cost-effective manufacturing of specific antigenic components, such as Com1 and P1, which can be purified and formulated into vaccines [163]. Recombinant subunit vaccines offer the potential for a more targeted and consistent production process compared to traditional methods [105,162]. Several subunit vaccine candidates have been developed and evaluated in preclinical studies, targeting various antigenic components of *C. burnetii*. A subunit vaccine containing the outer membrane protein Com1, formulated with a TLR triagonist adjuvant, demonstrated strong IgG2c-skewed antibody responses in mice but only conferred partial protection against *C. burnetii* challenge compared to the whole-cell vaccine “Q-VAX” [138,143]. Another study by Fratzke et al. investigated a subunit vaccine incorporating six *C. burnetii* antigens combined with different TLR agonists as adjuvants [162]. Among the various formulations tested, only the vaccine containing TLR4, TLR7, and TLR9 agonists as tritagonists exhibited reduced reactogenicity while inducing protective immunity comparable to WCVs regarding bacterial burden reduction [162,164]. A phase I lipopolysaccharide (LPS I) of *C. burnetii* has been identified as a major virulence factor and a potential target for subunit vaccine development [165]. Immunization with purified LPS I has been shown to reduce *C. burnetii* loads in the spleens of mice following intraperitoneal challenge, although protection was less effective against aerosol challenge [29]. A synthetic peptide mimetic of LPS I conjugated to keyhole limpet hemocyanin (KLH) elicited transferable humoral responses in mice but displayed a lower protective efficacy than LPS I immunization [166]. In addition to protein and polysaccharide antigens, T-cell epitopes have been explored as potential components of subunit vaccines. A recent study evaluated a multi-epitope vaccine containing human leukocyte antigen (HLA) class II T-cell epitopes derived from *C. burnetii* antigens [167]. While the vaccine candidates induced non-reactogenic responses in a sensitized guinea pig model, the immune response in mice was predominantly directed against a single epitope, limiting its protective efficacy against the *C. burnetii* challenge [168]. Advances in antigen discovery and vaccine delivery systems have further expanded the possibilities for subunit vaccine development. Identifying novel immunogenic proteins through proteomics and immunoinformatics approaches has provided a broader range of potential vaccine targets [169]. Additionally, nanoparticle-based delivery systems and modern adjuvants have shown promise in enhancing the immunogenicity and efficacy of subunit vaccines [170]. Despite the progress made in subunit vaccine development, challenges remain in achieving optimal protection against *C. burnetii* infection. Selecting appropriate antigens, adjuvants, and delivery systems is crucial for eliciting robust and long-lasting immune responses [114,140]. Moreover, the lack of standardized animal models and correlates of protection hampers the evaluation and comparison of different subunit vaccine candidates [76]. In conclusion, subunit vaccines offer a targeted and potentially safer Q fever prevention approach than whole-cell vaccines. These vaccines aim to minimize reactogenicity while inducing protective immunity by focusing on specific antigenic components. However, further research is needed to identify the most promising antigen combinations, optimize vaccine formulations, and establish reliable correlates of protection. As subunit vaccine technology continues to evolve, it holds promise for developing effective and well-tolerated Q fever vaccines.

### 6.3. DNA- and RNA-Based Vaccines

DNA- and RNA-based vaccines represent innovative approaches to Q fever prevention, offering several advantages over traditional vaccine platforms. These vaccines deliver genetic material encoding specific *C. burnetii* antigens, allowing the host cells to express the antigens and stimulate an immune response [171,172].

DNA vaccines, consisting of plasmid vectors encoding immunogenic proteins, have shown promising results in preclinical studies [173]. Advances in delivery methods have significantly improved the immunogenicity of DNA vaccines. Electroporation, which uses electrical pulses to facilitate DNA uptake into cells, has enhanced cellular uptake and antigen expression, resulting in stronger T-cell responses than conventional delivery methods [174]. Additionally, codon optimization and more efficient promoter sequences have increased antigen expression levels, further boosting the immune response [175].

RNA-based vaccines, particularly self-amplifying mRNA constructs, have emerged as another promising platform for Q fever prevention. These vaccines offer several advantages, including rapid manufacturing, improved stability, and the ability to induce robust immune responses [176,177]. In preclinical studies, mRNA vaccines encoding Phase I antigens have demonstrated high seroconversion rates and the potential to cause long-lasting immunity [176].One key benefit of DNA- and RNA-based vaccines is their safety profile. Unlike live-attenuated or inactivated whole-cell vaccines, these vaccines do not contain infectious agents, reducing the risk of adverse reactions. Moreover, the ability to selectively include specific antigenic targets minimizes the potential for inducing harmful immune responses [176,178].

However, developing and implementing DNA- and RNA-based vaccines for Q fever remains challenging. While these platforms have shown promising immunogenicity in animal models, their efficacy in humans must be further evaluated [76]. Additionally, these vaccines’ long-term stability and storage requirements, particularly for RNA-based constructs, may pose logistical challenges in real-world settings [168].

Ongoing research focuses on optimizing vaccine design, delivery systems, and formulations to address these challenges. Novel adjuvants and advanced nanoparticle delivery vehicles are being explored to enhance the immunogenicity and stability of DNA- and RNA-based vaccines [179]. Furthermore, developing thermostable formulations and improved storage conditions could facilitate these vaccines’ widespread distribution and use [180].

In conclusion, DNA- and RNA-based vaccines offer a promising alternative to traditional Q fever vaccine approaches. These platforms aim to induce robust and long-lasting immune responses while minimizing the risk of adverse reactions by delivering specific antigenic targets and leveraging advanced delivery methods. As research progresses and clinical trials provide further insights into their efficacy and safety, DNA- and RNA-based vaccines may play an increasingly important role in preventing Q fever.

### 6.4. Multi-Epitope Vaccines

Multi-epitope vaccines represent an innovative strategy in Q fever vaccine development. They aim to provide broad protection by combining multiple immunogenic epitopes from various *C. burnetii* antigens. This approach allows for the rational design of vaccines that elicit targeted immune responses while minimizing potential adverse reactions [141]. Developing multi-epitope vaccines involves identifying and selecting highly conserved and immunogenic epitopes from different *C. burnetii* proteins. Advances in immunoinformatics and epitope mapping techniques have greatly facilitated this process, enabling the prediction of B-cell and T-cell epitopes likely to induce protective immunity [157,181]. By focusing on conserved epitopes, multi-epitope vaccines have the potential to protect against a wide range of *C. burnetii* strains [141,142].

One of the key advantages of multi-epitope vaccines is their ability to induce a balanced and comprehensive immune response. By incorporating epitopes that stimulate both humoral and cellular immunity, these vaccines can promote the generation of neutralizing antibodies and antigen-specific T cells, which are essential for the effective clearance of *C. burnetii* infection [181]. Moreover, including epitopes from multiple antigens can help overcome the limitations of single-antigen vaccines, which may be less effective due to antigenic variation or immune evasion mechanisms employed by the pathogen [141]. The design of multi-epitope vaccines often involves using bioinformatics tools and algorithms to optimize epitope selection and arrangement. These tools can help identify epitopes with high binding affinity to major histocompatibility complex (MHC) molecules, ensuring efficient presentation to T cells [182]. Additionally, the spatial arrangement of epitopes within the vaccine construct can be optimized to minimize junctional epitopes and enhance the overall immunogenicity [181].

Several studies have investigated the potential of multi-epitope vaccines for Q fever prevention. Xiong et al. [34] designed a multi-epitope vaccine incorporating T-cell epitopes from various *C. burnetii* antigens and evaluated its immunogenicity in mice. Although the vaccine induced antigen-specific responses, protection against the *C. burnetii* challenge was only observed in mice immunized with a cocktail of epitopes, highlighting the importance of epitope selection and combination. Another study by Peng et al. [166] reported the development of a multi-epitope vaccine containing B-cell and T-cell epitopes from outer membrane proteins, T4SS components, and phase I LPS. These vaccines constructs elicited robust humoral and cellular immune responses in mice and conferred protection against the *C. burnetii* challenge.

The manufacturing of multi-epitope vaccines presents both opportunities and challenges. Synthetic peptide synthesis allows for precise control over epitope sequences and purity, enabling the production of well-defined vaccine constructs [183]. However, the scalability and cost-effectiveness of peptide synthesis need to be considered for large-scale manufacturing. Alternatively, recombinant expression systems can produce multi-epitope fusion proteins, purified and formulated as subunit vaccines [184].

Despite the promising potential of multi-epitope vaccines, further research is needed to optimize their design and evaluate their efficacy in diverse populations. Selecting epitopes that provide broad coverage across different HLA alleles and *C. burnetii* strains is crucial for developing vaccines with global applicability [185]. Additionally, incorporating appropriate adjuvants and delivery systems can help enhance the immunogenicity and long-term protection elicited by multi-epitope vaccines [141].

In conclusion, multi-epitope vaccines represent a promising approach to Q fever prevention, offering the potential for rational design, targeted immune responses, and broad protection. By combining carefully selected epitopes from multiple *C. burnetii* antigens, these vaccines aim to induce a comprehensive and balanced immune response while minimizing the risk of adverse reactions. As our understanding of *C. burnetii* immunology and epitope mapping techniques continues to advance, multi-epitope vaccines may play an increasingly important role in the fight against Q fever.

### 6.5. Novel Delivery Systems and Adjuvant Formulations

Recent innovations in vaccine delivery systems have significantly enhanced the effectiveness of various Q fever vaccine platforms. Lipid nanoparticle formulations have demonstrated particular success, achieving enhanced stability at 2–8°C for up to 24 months while maintaining immunogenicity [186]. These delivery systems provide superior antigen presentation and cellular uptake, resulting in improved immune responses. Advanced adjuvant combinations have been developed to optimize immune responses while minimizing reactogenicity. Novel formulations incorporating specific immune modulators have achieved balanced activation of cellular and humoral immunity [187,188]. These developments have particular significance for subunit and multi-epitope vaccines, where appropriate adjuvant selection can substantially enhance immunogenicity [189]. Comparative analysis of vaccine platforms reveals distinct advantages and limitations in terms of manufacturing considerations, safety profiles, and immunogenicity. Subunit vaccines demonstrate advantages in manufacturing simplicity and quality control, while DNA and RNA vaccines benefit from standardized production processes [189]. Multi-epitope vaccines, though more complex in design, offer efficient large-scale production capabilities [190]. Each platform consistently demonstrates improved safety compared to traditional WCVs. Regarding immunogenicity and protection, subunit vaccines excel in generating targeted immune responses [175], while DNA vaccines offer prolonged antigen expression [76]. Multi-epitope vaccines provide comprehensive immune activation through strategic epitope combinations [187]. These distinct advantages enable tailored approach selection based on specific population needs and protection requirements.

The diversity of available vaccine platforms has significantly advanced Q fever prevention capabilities. Continued development and refinement of these approaches, combined with innovations in delivery systems and adjuvant formulations, promise further improvements in vaccine effectiveness and accessibility.

## 7. Human Vaccines: Q-VAX and Its Impact

“Q-VAX”, licensed in Australia since 1989, remains the only approved vaccine for Q fever in humans. Developed from the Henzerling Phase I strain of *Coxiella burnetii*, this formalin-inactivated whole-cell vaccine has demonstrated remarkable efficacy in high-risk populations, including abattoir workers, farmers, and veterinarians. Clinical trials prior to licensure revealed 100% protection over 18 months in vaccinated individuals and robust immune responses characterized by high seroconversion rates and lymphoproliferative activity [191] (Table 1). Despite its success, “Q-VAX’s use has been limited by reactogenicity concerns, particularly in individuals with prior exposure to *C. burnetii*. Adverse reactions such as severe erythema and induration at the injection site necessitated the introduction of mandatory pre-vaccination screening, combining skin tests and serological assessments [115]. These measures, while effective in minimizing risks, have added logistical challenges, delaying widespread implementation. Nevertheless, long-term surveillance has confirmed the vaccine’s sustained efficacy, with protection rates exceeding 94% over five years and minimal breakthrough infections. Overall, “Q-VAX” has demonstrated durable protective efficacy, but its widespread use has been hampered by the potential for severe adverse reactions and the cumbersome pre-screening process associated with vaccine distribution. As the only licensed vaccine available for human Q fever pre-exposure prophylaxis, “Q-VAX” was deployed in the 2011 Q fever outbreak in the Netherlands.

Economic analyses of “Q-VAX” vaccination programs in Australia have further highlighted its cost-effectiveness. By preventing occupational cases of Q fever, particularly among abattoir workers, the vaccine has demonstrated a favorable cost–benefit ratio, with significant reductions in healthcare costs and productivity losses. The integration of systematic pre-vaccination screening and vaccine delivery into occupational health programs has been pivotal in achieving these outcomes [192].

### Veterinary Vaccines: Reducing Transmission at the Source

In animal populations, licensed vaccines have been instrumental in controlling Q fever at its source. Veterinary vaccines, such as Coxevac, target small ruminants like goats and sheep, which are primary reservoirs of *C. burnetii*. These vaccines significantly reduce bacterial shedding during parturition, a critical period for environmental contamination, and lower abortion rates in infected herds [37]. Field studies have demonstrated Coxevac’s efficacy in various settings. For example, large-scale vaccination of goat herds during the Netherlands outbreak (2007–2010) resulted in a 92% reduction in bacterial shedding and a marked decline in human cases linked to infected livestock [193]. Despite these successes, challenges persist. Vaccination does not entirely eliminate bacterial shedding, and its effectiveness can be influenced by factors such as the timing of administration, age of the animals, and previous exposure to *C. burnetii*.

The experiences with “Q-VAX” and Coxevac underscore the importance of integrating human and veterinary vaccination programs into broader public health frameworks. The Netherlands outbreak highlighted the value of a coordinated response, where simultaneous vaccination of humans and livestock, alongside enhanced surveillance and biosecurity measures, successfully curtailed disease spread. These efforts exemplify the One Health approach, emphasizing the interconnectedness of human, animal, and environmental health in managing zoonotic diseases.

While licensed vaccines have significantly mitigated the impact of Q fever, ongoing challenges demand continued innovation. Efforts are underway to improve vaccine formulations to reduce reactogenicity, enhance long-term immunity, and expand vaccine stability under varied storage conditions. Additionally, developing next-generation serological assays could streamline pre-vaccination screening, reducing the logistical burden of current protocols [142]. The broader application of veterinary vaccines in endemic regions, coupled with targeted human vaccination campaigns for high-risk groups, holds promise for global Q fever control. Advances in manufacturing and delivery systems, including novel adjuvants and thermostable formulations, will be critical to overcoming barriers to widespread vaccine access [159].

Licensed Q fever vaccines have proven to be indispensable tools in reducing the disease burden among humans and animals. By addressing key challenges and leveraging integrated approaches, these vaccines can continue to play a central role in controlling Q fever and preventing future outbreaks. Sustained investment in research, coupled with policy-driven implementation strategies, will ensure their lasting public health impact [76].

## 8. Conclusions

One Health is an approach that recognizes that the health of people is closely connected to the health of animals and our shared environment. The successful development and deployment of licensed vaccines for Q fever mark a significant achievement in the global One Health approach. These vaccines have played a crucial role in reducing the burden of Q fever in both human and animal populations, providing insights into effective disease prevention strategies and underscoring the importance of integrated approaches to zoonotic disease control. Developing effective vaccines against Coxiella burnetii is reliant ona complex interplay between immunological understanding, technological advancement, and practical implementation. While current whole-cell vaccines, particularly “Q-VAX”, demonstrate significant protective efficacy, their limitations regarding reactogenicity and pre-screening requirements drive continued innovation in vaccine design. The progression from traditional whole-cell approaches to modern subunit and synthetic vaccines reflects a growing understanding of *C. burnetii* immunology and host–pathogen interactions. Advances in vaccine technology, including novel adjuvant systems, delivery platforms, and antigen design strategies, offer promising avenues for improvement. The successful development of next-generation vaccines requires a careful balance between immunogenicity and safety while addressing manufacturing scalability and cost-effectiveness. Integrating emerging technologies, particularly in molecular design and delivery systems, may overcome current limitations while maintaining robust protection. The complexity of *C. burnetii* infection, particularly in chronic cases, necessitates continued research into immunological mechanisms and vaccine-induced protection. Understanding the balance between protective immunity and pathological responses remains crucial for developing effective prophylactic and therapeutic vaccines. Future success depends on sustained collaborative efforts across disciplines, combining basic research with practical implementation considerations. As research progresses, the focus must remain on developing vaccines suitable for broad implementation without extensive pre-screening requirements. This goal requires careful consideration of safety profiles, manufacturing processes, and delivery systems. The potential impact of improved Q fever vaccines extends beyond human health to veterinary applications, offering comprehensive disease control and prevention approaches. The path forward in Q fever vaccine development demands continued innovation while maintaining practical considerations of implementation and accessibility. Success in this endeavor will significantly impact global health, reducing the disease burden and enhancing the preparedness for future challenges in both human and animal populations.

## Figures and Tables

**Figure 1 vaccines-13-00151-f001:**
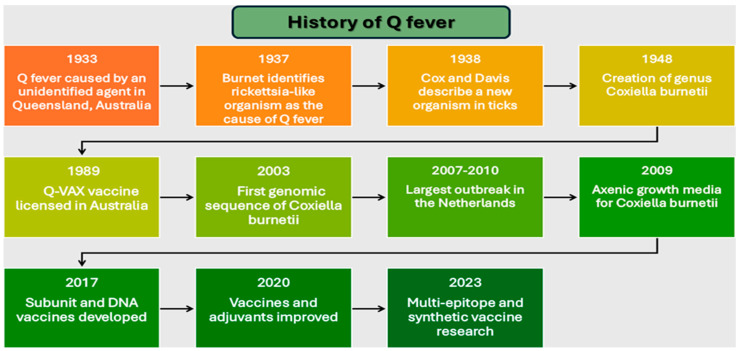
Timeline of major milestones in Q fever research and vaccine development (1933–2023).

**Figure 2 vaccines-13-00151-f002:**
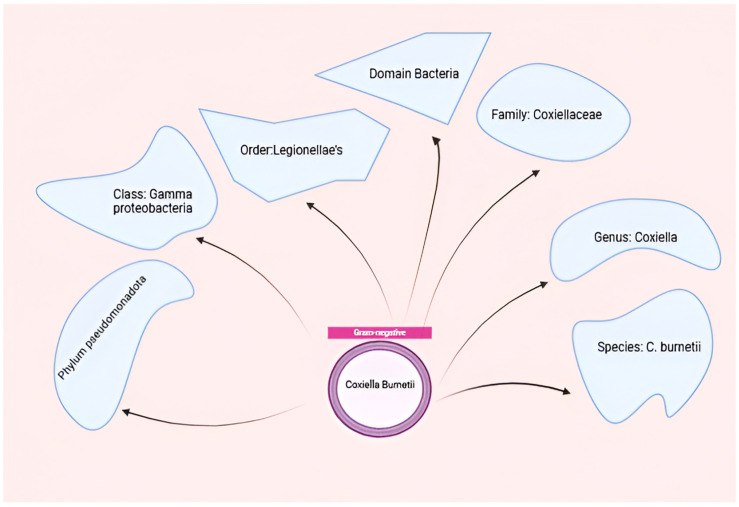
Taxonomy of *C. burnetii*. Figure was created using Biorender.com. Accessed on 29 December 2024.

**Figure 3 vaccines-13-00151-f003:**
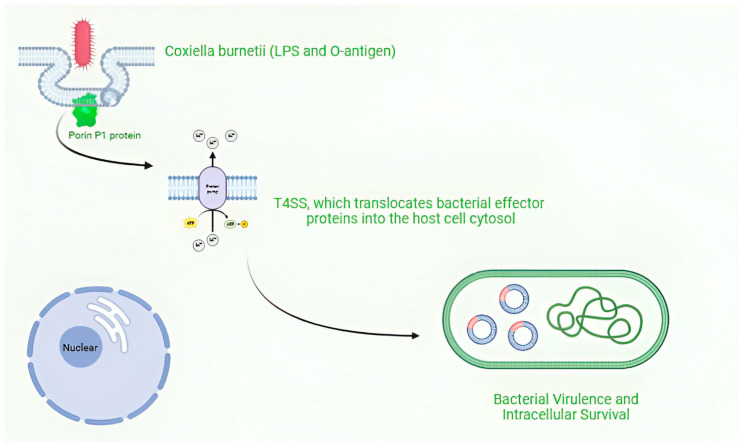
Intracellular survival and replication of Coxiella burnetii. Figure was created using Biorender.com. Accessed on 29 December 2024.

**Figure 4 vaccines-13-00151-f004:**
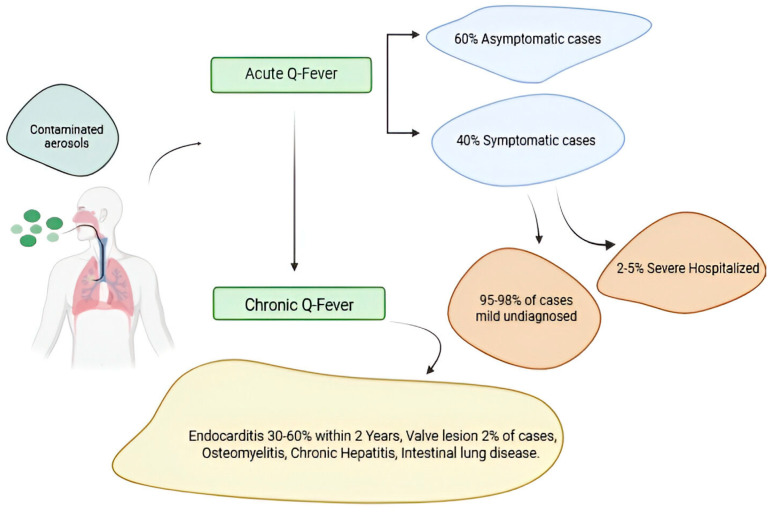
Q fever pathogenesis in acute and chronic forms. Figure was created using Biorender.com. Accessed on 29 December 2024.

**Table 1 vaccines-13-00151-t001:** Q fever vaccine characteristics.

Vaccine Type	Development Stage	Advantages	Limitations	Immune Response	Safety Profile	Manufacturing Considerations
Whole-Cell Vaccines (Q-VAX)	Licensed (Australia)	Proven efficacy (>95%)	Requires pre-screening	Strong humoral and cellular immunity	Local reactions common	Requires BSL-3 facilities
		Long-term protection	High reactogenicity	80–82% seroconversion	DTH in pre-sensitized individuals	Strict quality control
		Established manufacturing	Complex production	Robust T-cell response	Mandatory screening	Complex purification
Subunit Vaccines	Clinical Trials	Reduced reactogenicity	Variable protection	Targeted immune response	Improved safety profile	Simpler production
		Defined composition	Requires adjuvants	IgG2c-skewed antibodies	Minimal adverse events	Standard facilities
		Scalable production	Multiple doses needed	Lower cellular response	No pre-screening required	Consistent quality
DNA/RNA Vaccines	Preclinical/Early Clinical	No infectious material	Delivery challenges	Balanced immune response	Generally safe	Standard facilities
		Stable storage	Limited human data	Strong T-cell activation	No pre-screening required	Scalable process
		Precise design	Cost considerations	Sustained expression	Limited data available	Easier quality control
Multi-epitope Vaccines	Preclinical	Rational design	Complex design	Broad immune response	Promising safety profile	Consistent production
		Multiple targets	Limited data	Enhanced memory cells	Requires validation	High purity required
		Reduced reactogenicity	Manufacturing complexity	Multiple epitope targeting	No pre-screening required	Complex quality control

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
