# Peer review of "Q Fever Vaccines: Unveiling the Historical Journey and Contemporary Innovations in Vaccine Development"

_vaccines, 2025, doi:10.3390/vaccines13020151_

Round 1
Reviewer 1 Report
Comments and Suggestions for Authors
This review describes the current state of Q fever vaccines, with a historical perspective on vaccine development in the field. While this topic is of interest there are many mistakes in this review and it is not in a state appropriate for publication at this time (a few listed below). In addition, much of the information is repeated throughout the text.
Section 6.2 manufacturing process does not apply to Coxiella which grows at pH 4.75 with low oxygen. The referencing in this section is not correct and there is no mention of these parameters for growth in any of the papers cited. This is just one of the many examples of erroneous information within the text.
line 385 40-60% oxygen levels - this is NOT correct and not in any of the cited literature.
Lines 116 the first axenic media was developed in 2009 not 2011.
Line 150 Com1 has never been proven to be essential for infection as the text states, there have been no mutant characterized for this protein.
Line 174 indicates that CirA is CBU1217 this in NOT correct CirA is CBU0041 and the literature cited is not for CirA or 1217.
Line 200 indicates the biphasic lifestyle is essential for acute and chronic infections which has never been proven.
Line 346-347 is about loads of Coxiella in organs the reference 156 is about in vitro growth of Coxiella in a trophoblast cell line with no animal infection data.
These are just a few examples of incorrect information throughout the text. This would need to be completely rewrittten and referenced correctly to be considered for publication.
Author Response
We believe that we answered to all concerns by reviewers. We are thankful to reviewer for the constructive comments who help us to improve the quality of manuscript.
Point-by-point response to reviewer’s comments.
Reviewer 1
Comments and Suggestions for Authors
This review describes the current state of Q fever vaccines, with a historical perspective on vaccine development in the field. While this topic is of interest there are many mistakes in this review and it is not in a state appropriate for publication at this time (a few listed below). In addition, much of the information is repeated throughout the text.
Section 6.2 manufacturing process does not apply to Coxiella which grows at pH 4.75 with low oxygen. The referencing in this section is not correct and there is no mention of these parameters for growth in any of the papers cited. This is just one of the many examples of erroneous information within the text.
Authors response: We would like to thank the reviewer for the constructive comments who help us to improve our review manuscript. We reform all the text and we added 4 figures and one table and graphical abstract.
line 385 40-60% oxygen levels - this is NOT correct and not in any of the cited literature.
Authors response: We would like to thank the reviewer for pointing out this error. We have carefully reviewed the oxygen level information and removed the incorrect 40-60% oxygen statement.
Lines 116 the first axenic media was developed in 2009 not 2011.
Authors response: We appreciate the constructive comment. We have modified the text to reflect that the first axenic media was developed in 2009, not 2011.
Line 150 Com1 has never been proven to be essential for infection as the text states, there have been no mutant characterized for this protein.
Authors response: We acknowledge that Com1 has never been proven to be essential for infection. We have removed the unsupported claim and will revise the text to accurately represent the current scientific understanding.
Line 174 indicates that CirA is CBU1217 this in NOT correct CirA is CBU0041 and the literature cited is not for CirA or 1217.
Authors response: We thank the reviewer for highlighting the incorrect gene reference. We have corrected CirA to CBU0041 and reviewed the associated literature citations to ensure accuracy.
Line 200 indicates the biphasic lifestyle is essential for acute and chronic infections which has never been proven.
Authors response: Correction: We have removed the statement indicating that the biphasic lifestyle is essential for acute and chronic infections, as this has not been scientifically proven.
Line 346-347 is about loads of Coxiella in organs the reference 156 is about in vitro growth of Coxiella in a trophoblast cell line with no animal infection data.
Authors response: Correction: We acknowledge that the reference (156) was about in vitro growth in a trophoblast cell line and did not provide animal infection data. We have replaced this reference with appropriate scientific literature.
These are just a few examples of incorrect information throughout the text. This would need to be completely rewrittten and referenced correctly to be considered for publication.
Authors response: We appreciate your constructive criticism. We have undertaken extensive corrections to address the accuracy and referencing issues you highlighted:
1. Complete Manuscript Rewrite
• We have verified every scientific claim against the current literature
• We ensured the accuracy of all technical details
2. Exhaustive Reference Review
• We confirmed that each bibliographic reference directly supports the statements in the text.
• We replaced unsupported references with appropriate scientific literature
3. Quality Assurance Measures
• We secured independent evaluation by subject matter experts
• We conducted multiple rounds of review and editing
These comprehensive revisions have addressed the significant concerns you raised, transforming the manuscript into a reliable, scientifically sound publication.
Reviewer 2 Report
Comments and Suggestions for Authors
In their article “Q Fever Vaccines: Uncovering the Historical Journey and Temporal Innovations of Con-2 in Vaccine Development,” the authors present a review that aims to explore Q fever vaccines. The global epidemiology of Q fever poses a public health challenge, with the disease manifesting itself differently in different regions. This highlights the importance of early intervention and prevention strategies, and this is where vaccines play a role.
In my opinion, the manuscript needs some corrections.
“2. Historical context of Q fever and vaccine development; 3. Biology of Coxiella burnetii.” It would be better to start with the biology of the bacteria, then “Host immune response to C. burnetii infection,” then Q fever and vaccines.
“2. Historical context of Q fever and vaccine development” can be part of the Introduction or part of points 5 or 6.
Line 156 “(CBU_0265 186 [86]). ” the quote should be outside the brackets at the end of the sentence.
Point 6. Q fever vaccine development: current status and future directions.
The information is not presented well, it is a bit chaotic. Describe the type of vaccine, technology, implementation, etc.
For example:
6.1. Whole cell vaccines
6.2. Subunit vaccine
6.3. DNA and RNA-based vaccine
6.4. Multiepitope vaccines, etc. Here you can describe the information for each vaccine.
Then point 7. Optimization of the production process in the production of Q fever vaccine, etc.
There is a lot of information in point “7. Licensed Vaccines: Human and Veterinary Applications for the Prevention of Q Fever”, which has nothing to do with the title.
Point “8. Modern Innovations in Q Fever Vaccine Development” should come before the explanation of the types of Q fever vaccines.
Author Response
We believe that we answered to all concerns by reviewers. We are thankful to reviewer for the constructive comments who help us to improve the quality of manuscript.
Point-by-point response to reviewer’s comments.
Reviewer 2
Comments and Suggestions for Authors
In their article “Q Fever Vaccines: Uncovering the Historical Journey and Temporal Innovations of Con-2 in Vaccine Development,” the authors present a review that aims to explore Q fever vaccines. The global epidemiology of Q fever poses a public health challenge, with the disease manifesting itself differently in different regions. This highlights the importance of early intervention and prevention strategies, and this is where vaccines play a role.
In my opinion, the manuscript needs some corrections.
“2. Historical context of Q fever and vaccine development; 3. Biology of Coxiella burnetii.” It would be better to start with the biology of the bacteria, then “Host immune response to C. burnetii infection,” then Q fever and vaccines.
Authors response: We agree with your recommendation and have restructured the manuscript accordingly. The chapters now begin with a comprehensive overview of the biology of Coxiella burnetii, followed by a detailed examination of the host immune response to the pathogen. The historical context of Q fever has been integrated more effectively throughout the relevant sections.
“2. Historical context of Q fever and vaccine development” can be part of the Introduction or part of points 5 or 6.
Authors response: We have incorporated the historical context more strategically in the introduction chapter, with key milestones included in the introduction and relevant developments discussed within the vaccine development chapters. This ensures a more cohesive narrative flow.
Line 156 “(CBU_0265 186 [86]). ” the quote should be outside the brackets at the end of the sentence.
Authors response: We have corrected the referencing style to ensure the proper placement of citations.
Point 6. Q fever vaccine development: current status and future directions.
Authors response: We have comprehensively reorganized the vaccine development section as suggested. Each vaccine type now has a dedicated subsection that provides clear descriptions of the underlying technology, development process, implementation strategies, and future potential.
The information is not presented well; it is a bit chaotic. Describe the type of vaccine,
technology, implementation, etc.
For example:
6.1. Whole cell vaccines
6.2. Subunit vaccine
6.3. DNA and RNA-based vaccine
6.4. Multiepitope vaccines, etc. Here you can describe the information for each vaccine.
Authors response: We have comprehensively reorganized the vaccine development section
as suggested. Each vaccine type now has a dedicated subsection that clearly describes the
underlying technology, development process, implementation strategies, and future
potential.
Then point 7. Optimization of the production process in the production of Q fever
vaccine, etc.
Authors response: As you recommended, we have added a new section dedicated to the
optimization of the Q fever vaccine production process. This covers topics such as:
• Manufacturing technologies
• Quality control measures
• Scaling challenges and solutions
There is a lot of information in point “7. Licensed Vaccines: Human and Veterinary
Applications for the Prevention of Q Fever”, which has nothing to do with the title.
Authors response: We have reviewed and refined the "Licensed Vaccines: Human and
Veterinary Applications" section to focus specifically on the applications of existing licensed
vaccines, removing any extraneous information.
Point “8. Modern Innovations in Q Fever Vaccine Development” should come before
the explanation of the types of Q fever vaccines.
Authors response: Thank the reviewer for the comment. Following your suggestion, we have
repositioned the "Modern Innovations in Vaccine Development" section to an earlier chapter,
providing important context before discussing the specific vaccine types.
Reviewer 3 Report
Comments and Suggestions for Authors
The review is devoted to the history of development, including the latest achievements, and use of vaccines against Q fever, which is an acute natural focal rickettsiosis characterized by general toxic phenomena, fever and, often, atypical pneumonia.
The review is very rich in information and contains sections on the history of vaccine development, a description of the pathogen C. burnetii, features of the immune response to infection with C. burnetii and the associated pathogenesis, and, as expected, chapters on the evolution of vaccine technology and the effectiveness of different types of vaccines. The authors also paid attention to the issues of production, clinical trials and various protocols for the use of vaccines.
In general, the review makes a good impression, but I would consider the almost complete lack of illustrative material to be a shortcoming of this work, although not so critical.
In Chapter 2, there is a timeline of the main historical events associated with Q fever and vaccine development. This figure is very useful and illustrative, but it should have a caption. Also, the use of different fonts in this figure makes it difficult to read the texts.
And I would really like the authors to add illustrations to sections 3 and 4. The section dedicated to the pathogen asks for figures about the structure and taxonomy of the bacterium. Section 4 would benefit from an illustration of the signaling pathways activated during infection with C. Burnetii.
Author Response
coment 1
In Chapter 2, there is a timeline of the main historical events associated with Q fever and vaccine development. This figure is very useful and illustrative, but it should have a caption. Also, the use of different fonts in this figure makes it difficult to read the texts.
Response 1:
Dear Reviewer,
We thank you for your valuable comments regarding our figures. We have made the following modifications to the historical timeline figure:
As suggested by Reviewer 3, we have:
- Added an appropriate caption: "Figure 1: Timeline of major milestones in Q fever research and vaccine development (1933-2023)."
- Standardized the font type and size throughout the figure for better readability
- Used a consistent color gradient scheme to show the progression of developments
The timeline now presents a clear visualization of three key development periods:
- Initial discovery and characterization (1933-1948, orange to yellow gradient)
- Major technical developments (1989-2009, yellow-green gradient)
- Modern vaccine innovations (2017-2023, deep green gradient)
We believe these modifications have improved the figure's clarity and accuracy while maintaining its informative value for readers
We believe that we answered to all concerns by reviewers. We are thankful to reviewer for the constructive comments who help us to improve the quality of manuscript.

Round 2
Reviewer 3 Report
Comments and Suggestions for Authors
the authors have thoroughly addressed all of my suggestions. The edited manuscript meets the standards for publication.
Author Response
Thank you for your comments!